# Association between household food security and infant feeding practices among women with children aged 6–23 months in rural Zambia

Richard Bwalya[1]*, Chitalu Miriam Chama-Chiliba[1], Steven Malinga[2], Thomas Chirwa[3]

1 Institute of Economic and Social Research, University of Zambia, Lusaka, Zambia, 2 World Vision International, Uxbridge, United Kingdom, 3 World Vision Zambia, Lusaka, Zambia

* richkabemba@yahoo.com

**Data Availability Statement:** All relevant data are within the paper and Supporting information files.

**Funding:** The author(s) received no specific funding for this work.

## Abstract

Infant and young child feeding (IYCF) practices directly affect the nutritional status of children under two years of age, ultimately impacting their survival. However, ensuring that newborns and young children are fed according to the WHO-recommended practice has proven to be a challenge in many developing nations, especially in households that face food insecurity. This study aims to determine the association between IYCF practices and household food security's availability and access dimensions in rural Zambia. The study uses data from a cross-sectional survey of 2,127 mother-child pairs drawn from 28 rural districts in 8 out of the 10 Zambian provinces. Logistic regression analysis was used to examine the association of minimum dietary diversity, minimum meal frequency, and minimum acceptable diet with measures of household food security such as household dietary diversity score, and food insecurity experience scale, while controlling for confounding variables. The results show that children living in households classified as being food-secure based on the household dietary diversity score were significantly more likely to achieve appropriate feeding practices on all three IYCF measures, even after controlling for confounding factors. Notably, poor IYCF practices exist even in food-secure households, as most children in these households still need to receive a minimum acceptable diet. Although living in a household classified as food secure based on the access dimensions of household dietary diversity score and food insecurity experience scale is significantly associated with improvements in all three IYCF indicators even after controlling for confounding factors, the relationship does not hold for the availability measure of months of adequate household food provisioning. These findings highlight the need for targeting specific dimensions of household food security to solve child malnutrition, especially in rural areas. The focus should go beyond basic food availability, ensuring adequate diversity, and enhancing knowledge of appropriate feeding practices.

**Competing interests:** The authors have declared that no competing interests exist.

# 1 Introduction

Combating malnutrition in all its forms and food insecurity is one of the most significant global health challenges. The sustainable development goal (SDG) number two (2) emphasises this by committing all countries to end hunger, achieving food security and improving nutrition and promoting sustainable agriculture [1]. However, as of 2018, about one in three persons suffered from at least one form of malnutrition globally, while slightly more than 700 million people were exposed to severe levels of food insecurity [2]. The negative impacts of food insecurity on the wellbeing of children and adults have been well-documented [3–11]. Food insecurity among children is associated with increased risks of some birth defects, anaemia, low nutrient intake, cognitive problems, aggression and anxiety [12]. It is also associated with higher hospitalisation risks and poorer general health, chances of asthma, and behavioural problems [13–15].

Like many sub-Saharan African countries, Zambia has faced food insecurity and malnutrition challenges for decades. The 2019 Global Hunger Index (GHI), which is a composite of four indicators (undernourishment, child wasting, child stunting, and child mortality), lists Zambia among the six countries suffering from levels of hunger that are classified as alarming among the 117 countries that were ranked [16]. At 38.1, Zambia's GHI score was the fifth highest in 2019, only ranking better than countries that were in conflict or just out of conflict situations, such as Yemen (39.7), Chad (45.4), Central African Republic (53.7), and Madagascar (38.0). Regarding child malnutrition, Zambia is among the 34 countries with more than 20 percent stunting, which account for 90 percent of the global burden of malnutrition [17]. As of 2018, an estimated 35 percent of children under-five years of age were chronically under-nourished, 12 percent were underweight and 4 percent were wasted [18].

Infant and young child feeding (IYCF) practices directly affect the nutritional status of children under two years of age [19–21] and, ultimately, impact their survival [22–24]. For example, an estimated 14 percent of the deaths in children have been attributed to stunting and underweight, while wasting accounts for 12.6 percent [17]. As such, improving IYCF practices in children 0–23 months of age is critical to improved children's nutrition, health and development [25]. To this effect, eight core indicators were developed by the World Health Organisation (WHO). Four are recommended as a benchmark for evaluating the quality of IYCF practices [26]. These include the timely introduction of semi-solid or soft foods, minimum dietary diversity (MDD), minimum meal frequency (MMF), and minimum acceptable diet (MAD), which is a composite indicator of children that have met MMF and MDD. It is recommended that children be introduced to solid and semi-solid foods at the age of 6 months for the timely introduction of semi-solid or soft foods. For MDD, it is recommended that children aged 6–23 months be fed foods from four or more food groups out of seven groups daily. In addition, it is recommended that children should receive solid, semi-solid or soft food the minimum number of times or more; two times for breastfed children 6–8 months old, three times for those aged 9–23 months and four times for non-breastfed children aged 6–23 months [27].

Based on these metrics, many children in low and middle-income countries often do not meet the adequate complementary feeding requirements [2]. This is also the case for Zambia, where only 25.6 percent of the children 6–23 months received adequately diversified diets, 48.9 percent had the recommended meal frequency, while only 15.6 percent could be classified as having MAD [18]. The situation is even worse for rural children, where only 18.5 percent of the children (compared to 41.7 percent in urban areas) had adequately diversified diets, 46.4 percent (compared to 54.5 percent in urban areas) had the recommended meal frequency, while only 11.6 percent (compared to 24.6 percent) of the children could be classified as consuming MAD.

The persistent prevalence of inadequate IYCF practices requires more intensive and efficient efforts from different organisations to improve the situation, which can be achieved with a better understanding of the drivers of inappropriate IYCF practices and by identifying vulnerable groups [28, 29]. However, the available studies on the relationship between household food security and IYCF practices show conflicting results depending on the context. For instance, some studies in rural Bangladesh, Kenya and Ethiopia have shown a significant positive association between household food security and IYCF) practices [5, 10, 30], inappropriate IYCF practices have also been observed among children living in food-secure households [4]. Within Zambia, even though some studies have been conducted to characterise the IYCF practices [8, 9, 31] and identify the role of IYCF in child stunting [7, 32], none of these studies attempts to establish the role that household food security status plays in influencing IYCF practices, especially for rural areas.

The increased frequency of extreme weather events, such as prolonged droughts and flooding, are likely to compound the food security challenges for the rural populations that mainly depend on farming for their food needs as it reduces their asset base and consequently, the capacity to invest and adopt productivity-enhancing technologies. This will further aggravate the problem of food insecurity and inappropriate feeding practices among the poor [33]. The paucity of evidence emphasises the need to investigate how household food insecurity among rural households influences IYCF practices. A clear understanding of the association between household food insecurity and IYCF practices in rural settings is necessary to guide future interventions, such as nutrition-sensitive strategies to improve the nutritional status and the overall health of children in rural households. This study aims to assess the adequacy of IYCF practices among infants aged 6–23 months, examine the association between household food security dimensions and IYCF practices, and identify the key factors influencing IYCF practices in rural Zambian households.

## 2 Methods

### 2.1 Study design, sampling procedure, and sample size determination

The study was conducted in 32 Area Programmes (APs) where World Vision Zambia (WVZ) has been implementing interventions aimed at addressing problems of food insecurity. These interventions primarily focus on agriculture, water and sanitation, education, health, infrastructure development, livelihood and child protection, with the goal of transforming the lives of vulnerable children, women and families. An AP is a distinct geographical area where WVZ partners with local stakeholders to improve the well-being of children through multiple sector projects that target the root causes of issues negatively impacting children. These APs can vary in size, context, and population and often cover an entire administrative district. These 32 APs are distributed across 28 rural districts in 8 out of the 10 Zambian provinces. The major occupations of the population in these rural areas are mostly farming and trading. The data collection was part of a large programme evaluation conducted between April and June 2021.

The sample size for the evaluation was calculated using the sample calculation formula for cluster sampling, which allows for detecting statistical differences between surveys (S1 Data). The selection of 358 households in the sample was based on the minimum sample size required to detect a statistically significant difference between the proportions of the two surveys, using the indicator for the prevalence of stunting in children under five years of age at the AP level. The household selection used a 2-stage cluster sampling design with probability proportional to size (PPS) to ensure that larger zones had more sampled households than smaller ones. The sampling frame was developed from lists of zones, villages within the zones, along with the total population within each village. In the first stage of the sampling involved,

villages were randomly selected from the zones. In the second stage, the households interviewed were selected using the random walk method. Data on IYCF were collected using the caregiver questionnaire, with the mothers being the respondents according to WHO guidelines [25]. For this paper, a sub-sample comprising of 2,562 boys and girls aged 6–23 months is used.

## 2.2 Empirical strategy

This study assesses how household food security status influences complementary feeding practices for children aged 6–23 months, while controlling for confounding factors such as maternal characteristics and health-seeking behaviours, and household characteristics. The empirical strategy consists of running a series of binary logistic regressions, each representing a specific IYCF indicator in the form:

$$N_{ih} = \alpha + \beta_1 FS_{ih} + \beta_2 X_{ih} + \varepsilon \qquad (1)$$

Where $N_{ih}$ represents any of the three IYCF indicators such as MDD, MMF or MAD for child $i$ in household $h$; $FS_{ih}$ is a vector of household food security indicators that measure different dimensions of food security such as household dietary diversity score (HDDS), food insecurity experience scale (FIES), and months of adequate household food provisioning (MAHFP); and $X_{ih}$ is a vector of control variables which includes individual, household, demographic, socioeconomic and environmental characteristics; and $\varepsilon$ is the random error term. The term $\beta_1$ is the parameter of interest, representing the relationship between the household food security status of a given household and the mother's IYCF practices in that particular household.

The analysis involves running regressions for each of the three IYCF indicators (MMD, MMF, and MAD) as a function of the household food-insecurity status variables, controlling for confounding variables such as socio-economic status, household size, maternal education, sources of income, mother's decision-making score, health-seeking behaviours and exposure to maternal and child health interventions implemented by WVZ.

The binary logistic regression model provides a framework for assessing the likelihood of a mother in a given household feeding their child according to the recommended IYCF practices depending on the food security status of that particular household, while controlling for the confounding variables. The choice of the control variables is guided by similar studies that have been done both in Zambia [7–9, 32], and elsewhere [11, 34–36]. Eq (2) depicts the specification of the binary logistic regression model:

$$Z_i = \beta_0 + \sum (\beta_i X_{ki}) \qquad (2)$$

where $X_i$ represents a set of parameters that determine the feeding practices by the *ith* child's mother or caregiver. $Z_i$ represents the odds of a mother feeding the child according to the recommended IYCF practices or not, which is a dichotomous dependent variable (coded with 1 if the child is fed according to recommendations, and 0 otherwise). $\beta_0$ represents the model intercept and $\beta_1$ to $\beta i$ represents the coefficients of the explanatory variables, $X_1$ to $X_{ki}$.

$$P_i = \frac{e^{Z_i}}{1 + e^{Z_i}} \qquad (3)$$

where $P_i$ denotes the likelihood of the $i$th child's mother feeding them according to the recommended IYCF practices and $(1 − P_i)$ is the likelihood of the child not being fed according to the recommended IYCF practices. The odds ($Y = 1$ against $Y = 0$) define the proportion of the likelihood of a child being fed according to the recommended IYCF practices *(Pi)* to the

likelihood of a child not being fed according to the recommended practices *(1 − $P_i$)*; that is, odds = $P_i / (1 − P_i)$. Using the natural logarithm, the prediction is represented in Eq (4):

$$L_i = In \left( \frac{P_i}{1 - P_i} \right) = Z_i \qquad (4)$$

whereby the value of:

$$P_i = \left( \frac{1}{1 + e^{-Z_i}} \right) \qquad (5)$$

$Z_i$ is also denoted as the logarithm of the odds ratio in relation to a child being fed according to the recommended IYCF practices, as portrayed in the regression Eq (6):

$$Z_i = \beta_0 + \beta_1 X_1 + \beta_2 X_2 + \beta_3 X_3 + \beta_4 X_4 + \cdots \beta_n X_n + u_i \qquad (6)$$

where $Z_i$ represents the IYCF practices indicators (i.e., MMD, MMF, and MAD), which are assigned a numeric value of 1 if the child is fed according to the IYCF recommendations and 0 if otherwise), $\beta_0$ is the vector of unknown parameters (intercept), and $u_i$ denotes the error term.

## 2.3 Variables

**2.3.1 Dependent variables.** Three out of the eight IYCF practice indicators are used as outcome variables. These include MDD, MMF and the composite indicator, MAD, which is derived from the first two indicators. The MDD is defined as the proportion of children aged 6–23 months who consumed foods from at least four out of the seven referenced food groups within a 24-hour period. The seven food groups are: (1) grains, roots and tubers, (2) legumes and nuts, (3) dairy products, (4) flesh foods (meat, fish, poultry and liver/organ meats), (5) eggs, (6) vitamin A-rich fruits and vegetables, and (7) other fruits and vegetables [27]. The MMF represents the proportion of breastfed and non-breastfed children aged 6–23 months who received solid, semi-solid or soft foods (including milk feeds for non-breastfed children), the minimum required number of times. This minimum number of times is set at two and three meals for breastfed infants aged 6–8 months and 9–23 months, respectively, and four meals for non-breastfed children aged 6–23 months [25]. The MAD is defined as the proportion of children aged 6–23 months who met both the minimum meal frequency and dietary diversity criteria during the previous day [25].

**2.3.2 Explanatory variables.** Three groups of potential explanatory variables were identified from available data and literature [5, 7–10, 28–32]. The first group of potential variables relates to household food security status and aims to capture the various dimensions of household food security as defined by the World Food Summit of 1996. According to this definition, food security exists "when all people, at all times, have physical and economic access to sufficient, safe, and nutritious food that meets their dietary needs and food preferences for an active and healthy life" [38]. This definition of encompasses four pillars: food availability, accessibility, utilisation, and stability [37]. Availability refers to the physical presence of adequate food, accessibility involves individuals' access to resources for obtaining suitable foods for a nutritious diet, utilisation means having sufficient energy and nutrient intake combined with good biological absorption of the food consumed, and stability entails consistent access without losing such access [38].

To address the multi-dimensional nature of household food insecurity, three household food security indicators are used: HDDS, FIES, and MAHFP. Following earlier studies [39], the availability dimension is measured using the months of adequate household food

provisioning (MAHFP) indicator, while the access dimension of household food security is measured using two indicators, namely household dietary diversity score (HDDS) and food insecurity experience score (FIES).

The HDDS indicator measures food consumption reflecting household access to various foods. It is also meant to reflect, in a snapshot form, the economic ability of a household to access a variety of foods [40]. When assessed from the multi-dimensional nature of household food security, the HDDS indicator captures the utilisation dimension of household food security [42]. It is measured as the number of foods or nutritionally significant food groups acquired over the survey reference period and ranges from 0 to 12. The percentage of households with low diet diversity is a dummy variable taking a value of one for households acquiring fewer than four food groups out of twelve food groups during the reference period, and zero otherwise [41].

The FIES is an experience-based measure of household or individual food security and is the second indicator used in monitoring SDG target 2.1. The indicator, prevalence of moderate or severe food insecurity on the FIES, recognises that there are many people who, while not "hungry" in the sense that they suffer physical discomfort caused by severe lack of dietary energy, may still be food insecure [2]. These people can access food to meet their current energy requirements, yet are uncertain that it will last and may be forced to reduce the quality and/or quantity of the food they eat to get by. This moderate level of severity of food insecurity can contribute to various forms of malnutrition and has severe consequences for health and well-being. The FIES indicator captures the accessibility and stability dimensions of household food security [42]. The indicator "proportion of households facing moderate or severe food insecurity according to the Food Insecurity Experience Scale Global Standard Scale (FIES-GSS)" is defined as the percent of households who experience moderate or severe food insecurity [43]. It is a dummy variable taking a value of one if the household experienced moderate or severe food insecurity and zero otherwise.

The third indicator, MAHFP, serves as a proxy measure of household food access [44], capturing the food availability and stability dimensions [42]. A standardised questionnaire was to compute the MAHFP indicator by determining the total number of months that the households could adequately provide food for themselves in the previous 12 months. For each household, MAHFP was calculated by subtracting the total number of months out of the previous 12 months that the household could not meet their food needs from 12 months (i.e., within a given year). Food availability (related to the duration of food shortage experienced by the household) was coded on an ordinal scale, with households reporting between 5 to 11 months of food shortage being classified as having very low food availability, those with 3 to 4 months of food shortage were classified as having low food availability, those who reported 2 months of food shortage were classified as having medium food availability, while those who reported between 0 to 1 month of food shortage were classified as having high food availability [36].

Control variables focusing on maternal characteristics and child-health seeking behaviours and household economic characteristics are included in the analysis. The maternal characteristics and child-health seeking behaviours include the proportion of households with women actively engaged in decision making, which is defined as the proportion of households with adult women aged 18–49 living with a husband or partner who has a household decision-making index score of 0.67 or higher [45]. Other variables included whether the mother had no formal education, the mean number of under-five children in the household, whether the mother attended growth monitoring programmes (GMP) the recommended number of times, whether the child's mother sought antenatal care (ANC) the recommended number of times, the child's age in months, and whether the household benefited from maternal and child health programmes implemented by WVZ (i.e., residing in areas where WVZ implemented MNCH

programmes) The household economic characteristics variables included household size (measured as the total number of persons in a particular household), poverty likelihood, which characterised the households' asset ownership and probability of living below the poverty line [46], whether the household had alternative sources of income, and whether the household had a regular source of income.

## 2.4 Informed consent

Data collection was conducted only among mothers and caregivers aged 18 years and above. Enumerators obtained written informed consent from the mothers and caregivers, as required by the IRB study approval. Ethical clearance was granted by the Zambia ERES Converge Ethics Committee (reference number 2021-April-008). Enumerators were trained to obtain informed consent in languages familiar to potential respondents. Informed consent forms were translated into languages commonly spoken in the study sites. The informed consent procedure explained the study's purpose, and participants expressed their consent by either signing a consent form or providing a thumbprint on the form.

# 3 Results, discussion, and conclusions

## 3.1 Results

### 3.1.1 Characteristics of mothers, index children, and households.
Table 1 presents the household food security status, socioeconomic characteristics of the mothers, and index

**Table 1. Household food security and socioeconomic characteristics of the mothers and index children.**

| Household/ Maternal/ Child characteristic (N = 2,127) | N | Mean/% |
|---|---|---|
| **Household food security status** | | |
| • HH is food secure based on HDDS (%) | 1,692 | 66.0 |
| • HH experienced moderate or severe food insecurity (FIES) (%) | 750 | 29.3 |
| • Household food security status based on MAHFP | | |
| • Very low (%) | 155 | 6.0 |
| • Low (%) | 730 | 28.5 |
| • Medium (%) | 660 | 25.8 |
| • High (%) | 1,017 | 39.7 |
| **Maternal characteristics and practices** | | |
| • Mother has high decision-making score (%) | 1,519 | 71.4 |
| • Mother has no formal education (%) | 189 | 7.5 |
| • Total number of under 5 children (mean) | 2,562 | 1.7 |
| • Child attends GMP as recommended (%) | 1,002 | 39.1 |
| • Mother attended 4 or more ANC visits (%) | 1,927 | 75.2 |
| • Child's age in months | | |
| • 6 to 11 months | 973 | 38.0 |
| • 12 to 17 months | 905 | 35.3 |
| • 18 to 23 months | 648 | 26.7 |
| **Household characteristics** | | |
| • Household size (mean) | 2,562 | 6.7 |
| • Poverty likelihood (mean) | 2,562 | 69.3 |
| • Alternative income sources | 1,016 | 40.1 |
| • HH has regular source of income | 1,039 | 40.6 |
| • Total number of children in household | 2,562 | 4.1 |
| • Household resides in an area with WVZ MNCH interventions | 2,562 | 64.8 |

children. Out of the 2,562 mother and child pairs initially consenting to participate in the study, only 2,127 pairs are included in the analysis. The remaining pairs were omitted due to missing data on some variables. Regarding household food security characteristics, 66 percent of households were classified as food secure based on the HDDS indicator. About one third (29.3 percent) of the households experienced moderate or severe food insecurity based on the FIES indicator. Additionally, 34.5 percent had very low (6.0 percent) or low (28.9 percent) food availability based on the MAHFP indicator.

The majority (71.4 percent) of the mothers had a high decision-making score, which is alternatively defined as the proportion of households with women actively engaged in decision-making). Only 7.5 percent of the mothers had no formal education. About 39.1 percent of the mothers took their children for growth monitoring and promotion (GMP) programmes the recommended number of times, while 75.2 percent reported seeking ANC the recommended four or more times during pregnancy with the index child. The results also show that the mean number of children under the age of five was 1.7. Disaggregation of the children by age shows that the majority (38.0 percent) were aged between 6 and 11 months, followed by those aged between 12 and 17 months (35.3 percent), with 26.7 percent of the children aged between 18 and 23 months.

Household economic status variables show that the mean household size for the sample was 6.7 persons, with a mean number of children at 4.1. Approximately, 69.3 percent of the sampled households were living below the poverty line. Regarding household income, 40.1 percent reported having alternative sources of income, while another 40.6 percent reported at least one household member earning a regular income. The results also show that 64.8 percent of the households resided in areas where WVZ implemented maternal and child health programmes, implying that they benefited in some way from these programmes.

**3.1.2 Appropriate complementary feeding practices.** Fig 1 presents the IYCF practices of the mothers using the three selected WHO indicators. The results show that 49.9 percent of the children met the MMF criteria. Disaggregation by age shows that more children in the 6–11 months group (54.5 percent) met the MMF criteria compared to 49.1 percent in the 12–17 months age group and 44.6 percent in the 18–23 months age category. Only about a quarter (25.6 percent) of the sampled children met the MDD criteria. Disaggregation by age shows

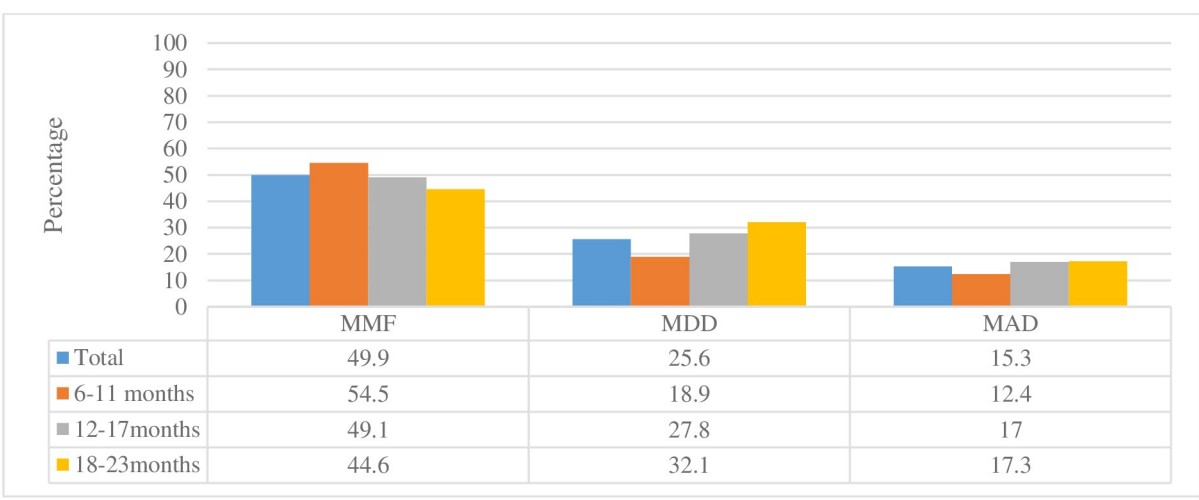

**Fig 1. Infant and young child practices of the mothers based on age categories using WHO selected indicator.**

that more children in the 18–23 months age group met the MDD criteria (32.1 percent) followed by those in the 12–17 months' age group (27.8 percent), with the lowest proportion meeting the MDD criteria being in the 6–11 months age group (18.9 percent). The results also show that only 15.3 percent of the children met the MAD criteria. Disaggregation by age shows that older children were more likely to meet the MAD criteria than younger children.

**3.1.3 Determinants of IYCF practices among mothers.** Table 2 shows the association between household food security indicators and IYCF indicators (Model 1), and how this relationship changes after accounting for explanatory variables (Model 2). The coefficients presented are the adjusted odds ratios from the binary logistic regressions for MMF, MDD, and MAD among children aged 6–23 months.

Living in a household without access to adequately diversified diets is associated with a reduced likelihood of children meeting the MDD, MMF, and MAD criteria. Compared to children living in households with adequately diversified diets (Table 2), those living in households without such diversification were 84.9 percent (AOR = 0.151, $p < 0.001$) less likely to meet the MDD criteria, 46.9 percent (AOR = 0.531, $p < 0.001$) less likely to meet the MMF criteria, and 79.8 percent (AOR = 0.202, $p < 0.001$) less likely to meet the MAD criteria even after controlling for confounding factors.

Compared to children living in households classified as moderately or severely food insecure based on FIES (Table 2), children living in households not classified as moderately or severely food insecure are 50.0 percent (AOR = 1.500, $p < 0.001$) more likely to meet the MFF criteria, and 33.3 percent (AOR = 1.334, $p < 0.10$) more likely to meet the MAD criteria even after controlling for confounding factors. However, being classified as moderately or severely food-insecure based on FIES does not significantly influence the likelihood of meeting the MDD criteria before or after controlling for confounding factors, despite the regression coefficients having the correct signs.

Living in a household classified as food insecure based on MAHFP is significantly associated with some IYCF indicators. For example, compared to those children living in households classified as having high food availability based on MAHFP, children living in households classified as having medium food availability were 24.8 percent (AOR = 0.752, $p < 0.001$) less likely to meet the MMF criteria even after controlling for confounding factors. Furthermore, although living in a household classified as having medium food availability reduced the likelihood of meeting the MAD criteria by 31.5 percent (AOR = 0.685, $p < 0.001$) when compared to living in households classified as having high food availability, this relationship became statistically insignificant once the control variables were introduced.

Other significant factors that influence whether a child follows recommended IYCF practices are the mother's decision-making index, education level, adherence to growth monitoring promotion and antenatal care recommendations and the child's age. For instance, compared to children with mothers having high decision-making scores, children whose mothers had a low decision-making index were 42.8 percent (AOR = 0.572, $p < 0.001$) less likely to meet the MDD criteria and 37.5 percent (AOR = 0.625, $p < 0.001$) less likely to meet the MAD criteria. Similarly, even though having a mother with no formal education does not significantly influence the MMF and MAD criteria, it is associated with a 43.5 percent (AOR = 0.565, $p < 0.001$) reduction in the likelihood of meeting the MDD criteria. Attending GMP and ANC according to recommendations were also associated with an increased likelihood of meeting the MDD and MAD criteria. For example, compared to children who were not taken for GMP as recommended, children taken for GMP based on the recommendations were 61.9 percent (AOR = 1.619, $p < 0.001$) and 65.5 percent (AOR = 1.655, $p < 0.001$) more likely to meet the MMF and MAD criteria respectively. On the other hand, compared to children whose mothers attended ANC the recommended four or more times, children whose mothers did not attend

**Table 2. Factors influencing the under–five children's nutrition status in Zambia.**

| Category | Variables | Statistic | MDD | | MMF | | MAD | |
|---|---|---|---|---|---|---|---|---|
| | | | Model 1 | Model 2 | Model 1 | Model 2 | Model 1 | Model 2 |
| Household food security status | HH is classified as being food insecure based on HDDS | B | -1.932 | -1.888 | -0.605 | -0.633 | -1.696 | -1.599 |
| | | (SE) | 0.144 | 0.168 | 0.087 | 0.101 | 0.178 | 0.200 |
| | | Exp (B) | 0.145*** | 0.151*** | 0.546*** | 0.531*** | 0.183*** | 0.202*** |
| | HH is classified as not being moderately or severely food insecure based on FIES | B | 0.108 | 0.074 | 0.490 | 0.406 | 0.302 | 0.288 |
| | | (SE) | 0.117 | 0.139 | 0.094 | 0.110 | 0.144 | 0.169 |
| | | Exp (B) | 1.114 | 1.077 | 1.632*** | 1.500*** | 1.353** | 1.333* |
| | HH is considered as having very low food availability based on MAHFP | B | -0.222 | -0.044 | -0.275 | -0.204 | -0.379 | -0.166 |
| | | (SE) | 0.035 | 0.272 | 0.181 | 0.206 | 0.286 | 0.315 |
| | | Exp (B) | 0.801 | 0.957 | 0.759 | 0.815 | 0.685 | 0.847 |
| | HH is considered as having low food availability based on MAHFP | B | 0.136 | 0.260 | -0.060 | 0.001 | 0.036 | 0.146 |
| | | (SE) | 0.120 | 0.140 | 0.102 | 0.117 | 0.138 | 0.158 |
| | | Exp (B) | 1.146 | 1.297* | 0.942 | 1.000 | 1.036 | 1.157 |
| | HH is considered as having medium food availability based on MAHFP | B | -0.054 | 0.101 | -0.297 | -0.284 | -0.378 | -0.254 |
| | | (SE) | 0.123 | 0.141 | 0.103 | 0.116 | 0.150 | 0.168 |
| | | Exp (B) | 0.948 | 1.106 | 0.743** | 0.752** | 0.685** | 0.776 |
| Maternal characteristics and practices | Mother has low decision-making score | B | | -0.558 | | 0.158 | | -0.470 |
| | | (SE) | | 0.132 | | 0.102 | | 0.153 |
| | | Exp (B) | | 0.572*** | | 1.172 | | 0.625** |
| | Mother has no formal education | B | | -0.571 | | -0.093 | | -0.435 |
| | | (SE) | | 0.259 | | 0.180 | | 0.310 |
| | | Exp (B) | | 0.565** | | 0.912 | | 0.647 |
| | Total number of under 5 children | B | | -0.051 | | 0.038 | | 0.055 |
| | | (SE) | | 0.074 | | 0.062 | | 0.085 |
| | | Exp (B) | | 0.950 | | 1.039 | | 1.056 |
| | Child attends GMP as recommended | B | | 0.267 | | 0.482 | | 0.504 |
| | | (SE) | | 0.175 | | 0.146 | | 0.203 |
| | | Exp (B) | | 1.306 | | 1.619** | | 1.655** |
| | Mother did not attended 4 or more ANC by skilled provider as recommended | B | | 0.080 | | -0.428 | | -0.434 |
| | | (SE) | | 0.130 | | 0.109 | | 0.162 |
| | | Exp (B) | | 1.084 | | 0.652*** | | 0.648** |
| | Child is aged 6–11 months | B | | -0.449 | | 0.684 | | 0.034 |
| | | (SE) | | 0.201 | | 0.169 | | 0.230 |
| | | Exp (B) | | 0.638** | | 1.981*** | | 1.035 |
| | Child is aged 12–17 months | B | | -0.089 | | 0.261 | | 0.120 |
| | | (SE) | | 0.136 | | 0.118 | | 0.157 |
| | | Exp (B) | | 0.915 | | 1.298** | | 1.127 |
| Household characteristics | Household size | B | | 0.064 | | -0.007 | | 0.001 |
| | | (SE) | | 0.021 | | 0.018 | | 0.024 |
| | | Exp (B) | | 1.066** | | 0.993 | | 1.001 |
| | Poverty likelihood | B | | -0.008 | | -0.004 | | -0.006 |
| | | (SE) | | 0.002 | | 0.002 | | 0.003 |
| | | Exp (B) | | 0.992*** | | 0.996** | | 0.994** |
| | HH has alternative income sources | B | | 0.261 | | 0.053 | | 0.128 |
| | | (SE) | | 0.116 | | 0.098 | | 0.134 |
| | | Exp (B) | | 1.298** | | 1.055 | | 1.137 |
| | HH has at least one member earning a regular income | B | | 0.017 | | 0.103 | | 0.115 |
| | | (SE) | | 0.115 | | 0.096 | | 0.132 |
| | | Exp (B) | | 1.017 | | 1.108 | | 1.122 |
| | Total number of children in the household | B | | -0.058 | | 0.030 | | -0.120 |
| | | (SE) | | 0.073 | | 0.062 | | 0.071 |
| | | Exp (B) | | 0.944 | | 1.030 | | 0.887* |
| | Household benefitted from WVZ maternal and child health programmes | B | | 0.081 | | 0.009 | | 0.038 |
| | | (SE) | | 0.116 | | 0.263 | | 0.134 |
| | | Exp (B) | | 1.085 | | 0.706 | | 1.039 |
| | Constant | B | -0.712 | -0.563 | -0.038 | -0.348 | -1.471 | -1.482 |
| | | (SE) | 0.129 | 0.310 | 0.109 | 0.263 | 0.156 | 0.359 |
| | | Exp (B) | 0.491 | 0.569* | 0.963 | 0.706 | 0.230*** | 0.227*** |

*(Continued)*

**Table 2.** (Continued)

| Category | Variables | Statistic | MDD | | MMF | | MAD | |
|---|---|---|---|---|---|---|---|---|
| | | | Model 1 | Model 2 | Model 1 | Model 2 | Model 1 | Model 2 |
| | *2 Log likelihood* | | 2,581 $\chi^2 = 273.026$, $df = 5$, $p<0.001$ | 2,008.244 $\chi^2 = 320,891$, $df = 17$, $p<0.001$ | 3,437 $\chi^2 = 113.894$, $df = 5$, $p<0.001$ | 2,743.395 $\chi^2 = 135.765$, $df = 17$, $p<0.001$ | 2,035 $\chi^2 = 160.563$, $df = 5$, $p<0.001$ | 1,625.881 $\chi^2 = 172.665$, $df = 17$, $p = 0<0.001$ |
| | *$R^2$ (Nagelkerke)* | | 0.152 | 0.214 | 0.058 | 0.084 | 0.106 | 0.140 |
| | *Hosmer & Lemeshow test* | | $p = 0.274$ | $p = 0.209$ | $p = 0.441$ | $p = 0.679$ | $p = 0.811$ | $p = 0.961$ |
| | *Classification accuracy (%)* | | 74.4 | 75.2 | 58.7 | 61.3 | 84.7 | 84.4 |
| | *N* | | 2,562 | 2,562 | 2,562 | 2,562 | 2,562 | 2,562 |

Note: Standard errors in parenthesis

*$p < 0.10$,

**$p < 0.05$,

***$p < 0.01$;

one tailed t–test

ANC the recommended number of times were 34.8 percent (AOR = 0.652, $p < 0.001$) and 35.2 percent (AOR = 0.648, $p < 0.001$) less likely to meet the MMF and MAD criteria respectively. The results also show that the child's age significantly influences the mother's IYCF practices. For instance, compared to children aged 18–24 months, children aged 6–11 months were 36.2 percent (AOR = 0.638, $p < 0.001$) less likely to meet the MDD criteria, but 98.1 percent (AOR = 1.981, $p < 0.001$) more likely to meet the MMF criteria. Similarly, compared to children aged 18–24 months, children aged 12–17 months were 29.8 percent (AOR = 1.298, $p < 0.001$) more likely to meet the MMF criteria.

The results also show that a one-person increase in household size increased the likelihood of a child meeting the MDD criteria by 6.6 percent (AOR = 1.066, $p < 0.001$), though the association with the other IYCF indicators was statistically insignificant. A one-percent increase in the likelihood of living below the national poverty line was associated with a 0.8 percent (AOR = 0.992, $p < 0.001$), 0.4 percent (AOR = 0.996, $p < 0.001$), and 0.6 percent (AOR = 0.994, $p < 0.001$) reduction in the likelihood of children meeting the MDD, MMF and MAD criteria, respectively. Finally, compared to children living in households where the mothers reported not having alternative sources of income, living in a household where the mother had alternative sources of income increased the likelihood of meeting the MDD criteria by 29.8 percent (AOR = 1.298, $p < 0.001$).

## 3.2 Discussion

In general, the findings indicate that the various dimensions of household food security variables are significantly associated with the three IYCF indicators, although the associations depend on the dimension of food security under consideration. Living in a household classified as food secure based on HDDS is associated with improvements in all three IYCF indicators, but this is not the case for the remaining household food security indicators. For example, even though living in a household classified as moderately or severely food insecure based on FIES is significantly associated with MMF and MAD, even after controlling for confounding factors, it is not significantly associated with MDD. Similar to the findings on FIES, even though living in a household classified as food insecure based on MAHFP is significantly associated with MMF and MAD, it is not significantly associated with MDD. At the same time, the association between MAHFP and MAD also becomes insignificant after controlling for confounding factors.

Regarding the adequacy of IYCF practices in rural Zambia, the study shows that about half of the sampled children (49.9 percent) achieved the minimum meal frequency, 25.6 percent achieved minimum dietary diversity, while only 15.3 percent achieved minimum acceptable diets. These findings are slightly better than the national averages for rural Zambia, which estimate minimum meal frequency at 41.7 percent, minimum dietary diversity at 18.5 percent, and minimum acceptable diets at 11.6 percent [18]. A plausible explanation is that the study concentrated on areas receiving interventions aimed at improving IYCF practices, among other areas, from WVZ for about four years. However, disaggregating the APs based on whether they benefited from WVZ MNCHN interventions or not shows no significant difference in child feeding practices between those in areas receiving MNCH interventions and those that did not receive these interventions.

The results also show that even though children aged 6–11 months constitute the largest proportion who receive MMF, they also constitute the smallest proportion receiving MDD, implying that they also fare very poorly relative to the other age groups with regard to MAD. Similar findings from other low income countries in Africa [30] also show that although large proportions of children receive meals at the appropriate frequency, these meals tend to be limited in the variety of foods offered. The low percentage of children receiving adequately diversified foods is reflected in the low percentage of children receiving minimum acceptable diets (15.3 percent), leaving most children susceptible to malnutrition.

Similar to earlier studies from Bangladesh [10], Kenya [30, 47], and Ethiopia [5, 48] that have shown a significant positive association between household food security and IYCF practices, our study also shows that household food security significantly correlates with MMF, MDD, and MAD [4]. However, unlike the earlier studies that only focus on the association between household food security and child feeding practices, this study goes a step further by assessing how two different dimensions of household food security, namely access and availability, influence child feeding practices. In general, the results show that whereas the availability dimensions of household food security (MAHFP) significantly influence the MMF criteria of the IYCF practices, the access dimensions (HDDS and FIES) influence both the MMF and MDD dimensions, and subsequently, the MAD dimensions of IYCF even after controlling for confounding variables. The lower likelihood of children in households with inadequate access to food is mainly due to the failure to meet the MDD criteria in the IYCF practices. This implies that household food security status has to improve beyond the basic level of availability also to include improved access to translate into mothers providing acceptably diversified diets for the children and consequently MAD.

Similar to Agbadi [4], the results show that poor IYCF practices also exist even in food secure households as most of the households did not meet MAD criteria, implying that there are other factors other than household food security that influence IYCF practices. The results show that other factors such as high decision-making score for women relative to men, access to formal education, poverty levels and having access to alternative income sources, attendance of growth monitoring programmes and antenatal care for pregnant women that mediated on the relationship between household food security and IYCF practices. These findings are explained by the fact that even in households that have adequate access to food overall, individual household members may be food-insecure. Disparities within the household could arise because of differences in nutrient needs, for example during pregnancy, lactation and infancy, or because of inequality in distribution among members of the household [39]. In such instances, household feeding priorities and altruistic behaviour or food buffering of children regardless of household food security-insecurity status may be an important moderator of the link between household food security and child diet. As such, even children living in food insecure households, but with mothers who have high-decision making capabilities or knowledge

are likely to receive diversified diets as mothers make deliberate decisions to cater to these children's needs.

## 3.3 Strengths and limitations of the study

Compared to similar studies conducted in Zambia, a major strength of this study is the large sample size, which increases the power of analyses, as well as the wide geographical coverage that makes it generalisable to Zambia's rural settings. However, like any other observational study, this one also has some limitations. Unlike similar studies that have used diaries or direct observation of child feeding practices and the household's food consumption patterns, this study relies on maternal recall, which in some cases may introduce biases in the data. However, the time lag between dietary intake and reporting is comparatively small, 24 hours for the child and seven days for the household. The reliance on secondary data also implies that some variables that have been shown to influence child feeding practices were omitted as they were not part of the original survey design. For instance, detailed information on child eating habits as well as reasons why mothers did not give specific foods to children could not be explored due to data limitations. Similarly, whereas investigating the association between maternal education, child feeding practices and their impact on child nutrition would have provided more insights into the importance of maternal formal education on child nutrition outcomes, this also could not be done due to data limitations. However, these limitations present important avenues for future research in identifying barriers to optimal infant feeding practices in low-income settings.

## 3.4 Conclusions

This study aimed to assess the adequacy of IYCF practices for infants aged 6–23 months in selected parts of rural Zambia, identify key factors influencing these practices, and assess the association between household food security dimensions (availability and access) and IYCF practices. The findings of the study reveal several importants insights. Despite about half of the sample children receiving the recommended number of meals, only about a quarter of the them were receiving adequately diversified diets. It is particularly concerning that inadequately diversified diets is more prevalent among children 6–11 months. This highlights the need for targeted interventions to address poor IYCF practices in rural Zambia, with a specific focus on enhancing dietary diversity among infants and children aged 6–11 months.

The study underscores that different dimensions of food security have varying effects on different aspects of IYCF practices. Household food security should extend beyond the mere availability of food within households. It should encompass improved access to diverse foods for all household members. This dual emphasis is essential for mothers to provide acceptably diversified diets to their children, thereby ensuring better IYCF practices.

Household dietary diversity is significantly associated with an increased likelihood of children meeting the MDD criteria. Therefore, policies and interventions aimed at improving child nutrition should not only focus on increasing cereal production but also prioritise promoting household dietary diversity through crop diversification. A diverse range of foods at the household level is crucial for achieving better IYCF practices. Finally, the results show that poor IYCF practices persist even in food-secure households. This highlights the need for policies and interventions that address household food security to be complemented by efforts to improve mothers' knowledge regarding appropriate feeding practices. Special attention should be given to educating mothers on providing adequately diversified diets for children aged between 6 to 11 months.

Achieving better IYCF practices requires not only enhancing the availability of food within households but also improving access to diverse foods and empowering mothers with the knowledge to provide balanced diets to their children.

## Supporting information

**S1 Data.**
(SAV)

## Acknowledgments

Special thanks to all the respondents who participated in the evaluation study for providing information that was used for writing this paper.

## Author Contributions

**Conceptualization:** Richard Bwalya, Chitalu Miriam Chama-Chiliba.

**Data curation:** Richard Bwalya.

**Formal analysis:** Richard Bwalya.

**Writing – original draft:** Richard Bwalya, Chitalu Miriam Chama-Chiliba.

**Writing – review & editing:** Richard Bwalya, Chitalu Miriam Chama-Chiliba, Steven Malinga, Thomas Chirwa.

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
