## [Decision Letter · Decision Letter 0]

21 Dec 2022

PONE-D-22-27270Association between household food security and infant feeding practices among women with children aged 6 – 23 months in rural ZambiaPLOS ONE

Dear Dr. Bwalya,

Thank you for submitting your manuscript to PLOS ONE. After careful consideration, we feel that it has merit but does not fully meet PLOS ONE’s publication criteria as it currently stands. Therefore, we invite you to submit a revised version of the manuscript that addresses the points raised during the review process.

Please address the concerns raised by reviewers point by point and submit a revised version of the manuscript.**********==============================

We look forward to receiving your revised manuscript.

Kind regards,

Prof Sajid Bashir Soofi

Academic Editor

PLOS ONE

Journal Requirements:

2. You indicated that you had ethical approval for your study. Please clarify whether minors (participants under the age of 18 years) were included in this study. If yes, in your Methods section, please ensure you have also stated whether you obtained consent from parents or guardians of the minors included in the study or whether the research ethics committee or IRB specifically waived the need for their consent.

Additional Editor Comments (if provided):

Please address the concern raised by reviewers point by point and submit a revised version of the manuscript.

Reviewers' comments:

Reviewer's Responses to Questions

**Comments to the Author**

1. Is the manuscript technically sound, and do the data support the conclusions?

Reviewer #1: Yes

Reviewer #2: Yes

2. Has the statistical analysis been performed appropriately and rigorously? 

Reviewer #1: I Don't Know

Reviewer #2: Yes

3. Have the authors made all data underlying the findings in their manuscript fully available?

Reviewer #1: Yes

Reviewer #2: Yes

4. Is the manuscript presented in an intelligible fashion and written in standard English?

Reviewer #1: Yes

Reviewer #2: Yes

5. Review Comments to the Author

Reviewer #1: Research fills a pertinent knowledge gap. It addresses an important issue of public health concern. Methodology is very robust. However to strengthen the quality of work weaknesses/limitations of the research should also be mentioned explicitly in the discussion section.

Reviewer #2: Association between house hold food security and infant feeding practices among women with children aged 6-23 months in rural Zambia.

The paper present overview of household food security and infant feeding practices. The paper is well written and presents interesting findings from population living in rural areas. The sample size is adequate and represents the population adequately.

Following are few suggestions:

Line 9 please change “proved” to proven

Line 19 remove “associated”

Line 55-56 structure of sentence, please change to “List Zambia among the six countries out of 117 countries, suffering from levels of hunger that are classified as alarming.

Line 211 please remove “that”

The study addresses the household food security and infant feeding practices in rural Zambia. While the study has an elegant design there are certain gaps in the study

1. The impact of the total number of children is not taken into account and is neither was discussed as a confounder

2. Position of the 6-23 months old child in the family. It is likely that the food availability was adequate but the number of children older than the index child might impact accessibility and availability of food for the index case

3. Did the investigator ask questions regarding alternative food that the child has access to. What percentage of these children were eating store bought or adult food.

4. How did the investigator measured child eating habit. Despite availability or accessibility of food, child eating behavior is also partially responsible for the overall health of the child. Did investigator provided a food diary to mothers/caregivers to measure the total amount of food intake in day/week or month.

6. PLOS authors have the option to publish the peer review history of their article (what does this mean?). If published, this will include your full peer review and any attached files.

Reviewer #1: No

Reviewer #2: **Yes: **Shaper Mirza

---

## [Author Response · Author response to Decision Letter 0]

7 Feb 2023

Response to Editor

Editor’s comment: 

Your ethics statement should only appear in the Methods section of your manuscript. If your ethics statement is written in any section besides the Methods, please move it to the Methods section and delete it from any other section. Please ensure that your ethics statement is included in your manuscript, as the ethics statement entered into the online submission form will not be published alongside your manuscript.

Response: 

We had added a section between the sections for conflict of interest and references, which was called “ethical statement”. This has been deleted as the same information was also contained in the “ethical statement” section of the methodology.

---

## [Decision Letter · Decision Letter 1]

17 Apr 2023

PONE-D-22-27270R1Association between household food security and infant feeding practices among women with children aged 6 – 23 months in rural ZambiaPLOS ONE

Dear Dr. Bwalya,

Thank you for submitting your manuscript to PLOS ONE. After careful consideration, we feel that it has merit but does not fully meet PLOS ONE’s publication criteria as it currently stands. Therefore, we invite you to submit a revised version of the manuscript that addresses the points raised during the review process.

We have received positive feedback from our reviewers on your submitted manuscript, and However, before we can proceed with the publication, there are a couple of important comments from one of the reviewers that need to be addressed. Please find the reviewer's comments attached to this email. We kindly request that you review them carefully and make the necessary revisions to your manuscript. Along with your revised submission, please provide a point-by-point response outlining how you have addressed each of the reviewer's concerns.

We look forward to receiving your revised manuscript.

Kind regards,

Sajid Bashir Soofi

Academic Editor

PLOS ONE

Journal Requirements:

Additional Editor Comments:

Dear Bwalya

Thank you for submitting the revised version of your manuscript. Our reviewers have carefully evaluated your revisions, and while the majority of the concerns have been addressed, one of the reviewers still has some minor comments that need to be considered.

We kindly request that you review the attached comments from the reviewer and make the necessary changes to your manuscript accordingly. Addressing these minor issues will help to further improve the quality and clarity of your work.

Once you have made the requested revisions, please resubmit your manuscript along with a point-by-point response to the reviewer's comments, detailing how you have addressed each concern. This will facilitate the evaluation process and expedite the final decision on your submission.

We appreciate your cooperation and understanding in this matter, and we look forward to receiving your revised manuscript. Should you have any questions or require further clarification, please do not hesitate to reach out to us.

Thank you once again for your valuable contribution to our publication.

Reviewers' comments:

Reviewer's Responses to Questions

**Comments to the Author**

1. If the authors have adequately addressed your comments raised in a previous round of review and you feel that this manuscript is now acceptable for publication, you may indicate that here to bypass the “Comments to the Author” section, enter your conflict of interest statement in the “Confidential to Editor” section, and submit your "Accept" recommendation.

Reviewer #1: All comments have been addressed

Reviewer #2: (No Response)

2. Is the manuscript technically sound, and do the data support the conclusions?

Reviewer #1: Yes

Reviewer #2: Yes

3. Has the statistical analysis been performed appropriately and rigorously? 

Reviewer #1: I Don't Know

Reviewer #2: Yes

4. Have the authors made all data underlying the findings in their manuscript fully available?

Reviewer #1: Yes

Reviewer #2: Yes

5. Is the manuscript presented in an intelligible fashion and written in standard English?

Reviewer #1: Yes

Reviewer #2: Yes

6. Review Comments to the Author

Reviewer #1: Research fills a pertinent knowledge gap. It addresses an important issue of public health concern. As per comments shared in previous review, strengths and limitations of the research have been added to the discussion section.

Reviewer #2: The manuscript is well written and provides in information on several aspects that could potentially impact child feeding practices. Below are few queries that should be addressed.

1. In discussion the author made a comment on the inequality in distribution of food among members and its impact on child feeding and MDD. However, this important point was not covered in the text. Statistical analysis taking into account the position of the child or the dietary habits of child was not addressed.

2. The study was conducted in an area that was supported by world vision Zambia. An important component of the program is educating mothers to improve feeding practises. Was this taken into account during analysis. It is likely that the education mothers are receiving through programs is sufficient for improving feeding practises and not having formal education is therefore not impacting feeding practises. I believe this is an important observation which was not taken into account during analysis.

3. Analysis showed that Young children (6-11 months old) were most significantly impacted by maternal feeding practises which makes sense because they cannot fend for themselves. Did author tried to investigate the association between maternal education, feeding practises and impact on child nutrition, I think this is where maternal formal education will have the most significant impact.

7. PLOS authors have the option to publish the peer review history of their article (what does this mean?). If published, this will include your full peer review and any attached files.

Reviewer #1: **Yes: **Sabeen Shah

Reviewer #2: **Yes: **Shaper Mirza

---

## [Author Response · Author response to Decision Letter 1]

20 Jun 2023

RESPONSE TO REVIEWER COMMENTS

Concern #1. In discussion the author made a comment on the inequality in distribution of food among members and its impact on child feeding and MDD. However, this important point was not covered in the text. Statistical analysis taking into account the position of the child or the dietary habits of child was not addressed.

How it has been addressed: 

On the question of statistical analysis taking into account the dietary habits of the child, we are unable to do that due to lack of data on the same. This was stated in the previous review and the limitation was acknowledged in the limitation section of the paper. With regard to the issue of the child’s position within the household, this variable cannot be added considering that we collect data on the index child, who happens to be the youngest child in the household. It is very similar to number of under 5 children in the household in that in a household with 5 children, the youngest child will be on position number 5. This similarity introduced multi-collinearity and thus the variable position of child is not included. We also do not discuss the variable in the text for number of children as it was statistically insignificant.

Concern #2. The study was conducted in an area that was supported by world vision Zambia. An important component of the program is educating mothers to improve feeding practises. Was this taken into account during analysis. It is likely that the education mothers are receiving through programs is sufficient for improving feeding practises and not having formal education is therefore not impacting feeding practises. I believe this is an important observation which was not taken into account during analysis.

How it has been addressed:

We have taken advantage of the fact that WVZ did not implement MNCHN programmes across all the intervention areas. To this effect, we added a new variable disaggregating the data between areas with MNCHN interventions and those without so as to assess whether feeding practices are different between these two areas. These changes have been effected in the introduction, results and discussion. However, the inclusion of this variable does not significantly influence the overall results. 

Concern #3. Analysis showed that Young children (6-11 months old) were most significantly impacted by maternal feeding practises which makes sense because they cannot fend for themselves. Did author tried to investigate the association between maternal education, feeding practises and impact on child nutrition, I think this is where maternal formal education will have the most significant impact.

How it has been addressed: 

The authors appreciate that investigating the association between maternal education, feeding practices and impact on child nutrition is very important. However, the focus of the paper was on association between household food security and infant feeding practices among women with children aged 6 – 23 months. While the impact of education on child feeding practices is investigated and discussed, we do not look at the three-way interaction between maternal education, feeding practices and child nutrition outcomes due to data limitations. However, this limitation has been acknowledged in the paper and included as a potential area for future studies.

---

## [Decision Letter · Decision Letter 2]

4 Sep 2023

PONE-D-22-27270R2Association between household food security and infant feeding practices among women with children aged 6 – 23 months in rural ZambiaPLOS ONE

Dear Dr. Richard Bwalya,

Thank you for submitting your manuscript to PLOS ONE. After careful consideration, we feel that it has merit but does not fully meet PLOS ONE’s publication criteria as it currently stands. Therefore, we invite you to submit a revised version of the manuscript that addresses the points raised during the review process.

We look forward to receiving your revised manuscript.

Kind regards,

Pradip Chouhan

Academic Editor

PLOS ONE

Journal Requirements:

Reviewers' comments:

Reviewer's Responses to Questions

**Comments to the Author**

1. If the authors have adequately addressed your comments raised in a previous round of review and you feel that this manuscript is now acceptable for publication, you may indicate that here to bypass the “Comments to the Author” section, enter your conflict of interest statement in the “Confidential to Editor” section, and submit your "Accept" recommendation.

Reviewer #2: All comments have been addressed

Reviewer #3: (No Response)

2. Is the manuscript technically sound, and do the data support the conclusions?

Reviewer #2: Yes

Reviewer #3: Yes

3. Has the statistical analysis been performed appropriately and rigorously? 

Reviewer #2: Yes

Reviewer #3: Yes

4. Have the authors made all data underlying the findings in their manuscript fully available?

Reviewer #2: (No Response)

Reviewer #3: Yes

5. Is the manuscript presented in an intelligible fashion and written in standard English?

Reviewer #2: Yes

Reviewer #3: Yes

6. Review Comments to the Author

Reviewer #2: The author made significant changes in the manuscript, which helped in clarifying some of the data discrepancies observed in previous draft. Results are now explained better and conclusions significantly improved and reflect results appropriately.

Reviewer #3: The study topic entitled "Association between household food security and infant feeding practices among women with children aged 6 – 23 months in rural Zambia" is a serious concern in present day as a developing country in Zambia. The manuscript finding and its relating suggestions would be helpful for policy makers, government to improve household food security and infant feeding practices among women with children aged 6 – 23 months in rural Zambia. Authors are suggested a minor corrections in their manuscript which are pointed below—

1. In Introduction part the line “Infant and young child feeding (IYCF) practices directly affect the nutritional status of children 63 under two years of age and, ultimately, impact their survival.” if this kind of fact is found in previously recent works then cite more (Line-62).

2. In discussion part “Similar to earlier studies, our study also shows that household food security significantly correlates with MMF, MDD, and MAD [4]” line-419 citation need for Similar to earlier studies for more strengthening the manuscripts.

3. Add Separate section with strength and limitation of the study.

4. Grammatical errors, tenses and syntax should be checked throughout the text.

7. PLOS authors have the option to publish the peer review history of their article (what does this mean?). If published, this will include your full peer review and any attached files.

Reviewer #2: **Yes: **Shaper Mirza

Reviewer #3: No

---

## [Author Response · Author response to Decision Letter 2]

11 Sep 2023

Response to reviewer’s comments

1. In Introduction part the line ―Infant and young child feeding (IYCF) practices directly affect the nutritional status of children 63 under two years of age and, ultimately, impact their survival. If this kind of fact is found in previously recent works, then cite more (Line-62).

• Response

- We have cited previous works showing the association between IYCF and child nutrition status as well as literature showing association between IYCF and child survival. We have also included statistics showing the proportion of child deaths arising from malnutrition. The reference section has also been adjusted accordingly

2. In discussion part ―Similar to earlier studies, our study also shows that household food security significantly correlates with MMF, MDD, and MAD [4] line-419 citation need for Similar to earlier studies for more strengthening the manuscripts.

• Response

- We have cited previous work that found significant association between household food security and child feeding practices. The references have also been adjusted accordingly

3. Add Separate section with strength and limitation of the study.

• Response

- The strengths and limitations of the current study were already included within the discussion section of the paper as a paragraph. We have put a section header within the discussion clearly labelled as “strengths and limitations of the study” to make it more visible

4. Grammatical errors, tenses and syntax should be checked throughout the text

• Response

- The whole document has been reviewed and grammatical errors, tenses and syntax revised accordingly

---

## [Editor Report · Decision Letter 3]

13 Sep 2023

Association between household food security and infant feeding practices among women with children aged 6 – 23 months in rural Zambia

PONE-D-22-27270R3

Dear Dr. Bwalya,

We’re pleased to inform you that your manuscript has been judged scientifically suitable for publication and will be formally accepted for publication once it meets all outstanding technical requirements.

Kind regards,

Pradip Chouhan

Academic Editor

PLOS ONE
---

## [Editor Report · Acceptance letter]

22 Sep 2023

PONE-D-22-27270R3 

Association between household food security and infant feeding practices among women with children aged 6 – 23 months in rural Zambia 

Dear Dr. Bwalya:

I'm pleased to inform you that your manuscript has been deemed suitable for publication in PLOS ONE. Congratulations! Your manuscript is now with our production department. 

Kind regards, 

on behalf of

Professor Pradip Chouhan 

Academic Editor

PLOS ONE